# How Is the Nociceptive Withdrawal Reflex Influenced by Increasing Doses of Propofol in Pigs?

**DOI:** 10.3390/ani14071081

**Published:** 2024-04-02

**Authors:** Alessandro Mirra, Ekaterina Gamez Maidanskaia, Olivier Louis Levionnois, Claudia Spadavecchia

**Affiliations:** Section of Anesthesiology and Pain Therapy, Department of Clinical Veterinary Medicine, Vetsuisse Faculty, University of Bern, 3012 Bern, Switzerland; ekaterina.gamez@unibe.ch (E.G.M.); olivier.levionnois@unibe.ch (O.L.L.); claudia.spadavecchia@unibe.ch (C.S.)

**Keywords:** pig, nociceptive withdrawal reflex, depth of anesthesia, propofol

## Abstract

**Simple Summary:**

The assessment of the depth of anesthesia in pigs is challenging. The nociceptive withdrawal reflex is a physiological, polysynaptic spinal reflex occurring in response to noxious stimulations, and the tracking of its thresholds has been shown to be possibly useful in evaluating the depth of anesthesia. In the present study, we investigated how increasing the dose of the anesthetic drug propofol influences the nociceptive withdrawal reflex thresholds in pigs. The nociceptive withdrawal reflex thresholds were continuously recorded while increasing the dose of propofol that was administered. Each animal underwent the same experimental protocol on three different days. We found that the nociceptive withdrawal reflex thresholds increased with an increasing dose of propofol, and that the same trend was present on the three experimental days. Our results suggest that the assessment of the nociceptive withdrawal reflex thresholds may complement the depth of anesthesia evaluation in pigs receiving propofol.

**Abstract:**

The nociceptive withdrawal reflex (NWR) is a physiological, polysynaptic spinal reflex occurring in response to noxious stimulations. Continuous NWR threshold (NWRt) tracking has been shown to be possibly useful in the depth of anesthesia assessment. The primary aim of this study was to describe how propofol modulates the NWRt over time in pigs. Five juvenile pigs (anesthetized three times) were included. An intravenous (IV) infusion of propofol (20 mg/kg/h) was started, and boli were administered to effect until intubation. Afterwards, the infusion was increased every ten minutes by 6 mg/kg/h, together with an IV bolus of 0.5 mg/kg, until reaching an electroencephalographic suppression ratio (SR) of between 10% and 30%. The NWRt was continuously monitored. For data analysis, the time span between 15 min following intubation and the end of propofol infusion was considered. Individual durations of propofol administration were divided into five equal time intervals for each pig (TI1–TI5). A linear regression between NWRt and TI was performed for each pig. Moreover, the baseline NWRt and slopes of the linear regression (b1) were compared among days using a Friedman Repeated Measures Analysis of Variance on Ranks. The NWRt always increased with the propofol dose (b1 = 4.71 ± 3.23; mean ± standard deviation). No significant differences were found between the baseline NWRt and the b1 values. Our results suggest that the NWRt may complement the depth of anesthesia assessment in pigs receiving propofol.

## 1. Introduction

General anesthesia is a complex combination of (at least) three outcomes, constituting what is generally defined as the “anesthesia triad” (unconsciousness, immobility, autonomic nervous system control) [1]. Despite the large use of the porcine model in translational medicine, appropriate objective methods to correctly evaluate the anesthetic level in pigs are still missing [2]. Quantitative methods, such as numerical variables derived from an electroencephalogram (e.g., bispectral index), have been investigated, but controversial results have been obtained [2]. Moreover, the specific monitoring of the brain function may not provide a complete assessment of the anesthetic triad.

One of the most commonly reported methods to guide anesthetic administration in experimental pigs is the evaluation of purposeful movements following noxious stimulations (e.g., claw clamp) [2,3,4]. Based on this reaction, the minimal alveolar concentration (MAC; for inhalant anesthetics) and the minimal infusion rate (MIR; for injectable anesthetics) have been commonly defined [5,6]. However, as the movement is assessed only as present/absent, it does not allow a fine assessment of the anesthetic depth.

The nociceptive withdrawal reflex (NWR) is a physiological, polysynaptic spinal reflex occurring in response to a noxious stimulation [7,8]. When the stimulation occurs at a peripheral nerve, afferent fibers are activated, the signal is transmitted to the spinal cord and the muscles involved in the withdrawal reaction are activated. The NWR has been primarily investigated to characterize nociceptive pathways and its pathophysiologic and pharmacologic modulation. Additionally, the NWR has also been shown to be a complementary adjunct in the anesthetic level assessment for different anesthetic regimens [9,10,11,12]. Contrary to the MAC/MIR, it allows a quantitative analysis of graded anesthetic-induced changes in sensory–motor processing [13].

While the effects of inhalant anesthetics on the NWR have been partially investigated, only scarce information is present for propofol, particularly on how it modulates the NWR in pigs [14,15,16,17,18]. Moreover, even if a good NWR threshold (NWRt) test–retest reliability has been demonstrated in humans [19,20], no information is present in pigs.

The primary aim of the present study was to describe how increasing the dose of propofol affects the NWRt over time in pigs; the secondary aim was to evaluate the reproducibility of the results if the same procedure is repeated on separate days.

We hypothesized that NWRt would rise with increasing doses of propofol, and that the results would not differ among days.

## 2. Materials and Methods

Ethical approval for the study (protocol number 32015) was obtained from the Committee for Animal Experiments of the Canton Bern. The ARRIVE guidelines were followed.

Five healthy pigs (phenotype Edelschwein) weighing 25.8 ± 1.8 kg (mean ± standard deviation), 10.4 ± 0.5-week-old and of mixed sex (four castrated males and one female) were included in a prospective experimental trial focusing on pharmacological enhancement of recovery from anesthesia. The pigs would be excluded and replaced in case of disease or severe anesthetic complications during the course of the experiment. Each pig was anesthetized three times on three separate days (Day 1, Day 2, Day 3). The same anesthetic protocol was used, ensuring a wash-out period of at least 36 h, while a different treatment was administered at the beginning of the recovery phase. At the end of the third experimental day, the pigs were euthanized using pentobarbital intravenously (IV; 150 mg/kg). For the purpose of the analysis presented here, only data collected during anesthesia were considered, such that the different treatments could be ignored and a total of fifteen anesthetic events evaluated. The sample size was calculated for the main investigation using the Resource Equation Approach [21], for analyzing the effect of methylphenidate on EEG-derived variables. Considering 3 groups, a minimum of 5 animals was required. This sample size also allows the detection of a significant doubling in the NWRt (Wilcoxon signed-rank test, one-tailed, effect size 1.8, alpha 0.05, power 0.90, SD 30%; G*Power 3.1.9.6 2020), which is in agreement with former studies on the NWR in unmedicated animals [22,23].

The animals were collected from the farm of origin and brought to the large animal facility of the Vetsuisse Faculty, University of Bern. At least two pigs at a time were kept in the facility, with visual and auditory contact continuously guaranteed. Each animal had at least 4 days of acclimatization before the start of the experiment, including a balanced general anesthesia for surgical placement of a subcutaneous jugular venous port on the second day of acclimatization. A fasting time of 6 h was applied before each anesthetic event; water was always available.

On the experimental day, the pigs were allowed to acclimatize for 30 min to the experimental room; then, they were placed in a purpose-made sling and instrumented while awake.

A six-lead wireless device (Televet 100, Engel Engineering Services GmbH, Heusenstamm, Germany) was used to continuously assess electrocardiographic activity.

A local anesthetic cream (EMLA, 5%, Anesderm, Pierre Fabre, Allschwil, Switzerland) was spread over both auricular veins and arteries, as well as over the coccygeal artery, and left to act for at least 45 min. Then, a venous and an arterial catheter (8 auricular, 5 coccygeal, 2 failure) were placed.

To determine the NWRt, a commercial device (PainTracker; Dolosys GmbH, Berlin, Germany) was used. Two surface electrodes were placed over the common dorsal digital nerve for transcutaneous electrical stimulation, and three surface electrodes were placed over the tibialis cranialis muscle on the same limb for the recording of electromyographic (EMG) activity (Figure 1).

Each stimulation consisted of five rectangular pulses of 1 ms duration delivered at 200 Hz. The initial stimulation intensity was set at 1 mA and was then automatically adjusted using a bracketing design [24] depending on the EMG response (positive or negative). Step size was set at 0.3 mA but increased to 0.5 mA when the intensity had changed in the same direction for the three last consecutive measurements. Stimulation was repeated every 10 s with 50% interval randomization. The time window considered for analysis of the EMG response (nociceptive reflex) was set between 40 and 140 ms following stimulus onset (NWR range), while the window between 130 and 10 ms prior to stimulation onset was assessed to quantify noise (noise range). Measurements were discarded when the EMG amplitude exceeded 15 microvolts (µV) within the noise range; in this case, the stimulus was repeated at the same intensity. The reflex was considered positive at an interval peak Z-score above 10, meaning that the difference between the maximum EMG amplitude within the NWR range and the mean EMG amplitude within the noise range had to be above 10-fold the standard deviation of the EMG amplitudes within the noise range [25]. The NWRt was automatically estimated after each stimulation through a logistic regression of the last 12 valid stimuli [25].

A paediatric RD SedLine electroencephalographic (EEG) sensor was placed over the frontal bone (Figure 2) as previously described, allowing recording of the EEG suppression ratio (SR) via the SedLine^®®^ monitor (Masimo Corp., Irvine, CA, USA) [26].

Briefly, the sensor was positioned on a transverse line over the frontal bone, keeping the rostral border on an imaginary line running between the lateral canthi of the eyes. The central CB (ground) and the caudal CT (reference) electrodes were placed on the mid-sagittal line. The sensor was repositioned if the impedance was above 10 kOhm. The SedLine display parameters were set at 10 μV/mm and 30 mm/s.

Recording of the NWRt started on the awake pig positioned in the sling. Then, 100% oxygen was supplied via a face mask for at least 5 min before propofol (Propofol 1% MCT, Fresenius Kabi AG, Switzerland) was administered to induce general anesthesia (always performed by the same operator: AM). A first IV bolus was administered (4–5 mg/kg) followed by smaller doses (0.5–1 mg/kg) repeated every 30–60 s, until endotracheal intubation was possible, with absence of jaw tone and tolerance to laryngeal stimulation (cuffed endotracheal tube, 6–7 mm internal diameter). Simultaneously with the first bolus, a continuous propofol IV infusion was started at 20 mg/kg/h. Following intubation, there was a 15 min wait to allow stabilization of the propofol plasmatic concentration. Then, propofol administration was increased by 6 mg/kg/h together with an additional IV bolus of 0.5 mg/kg. Thereafter, the same increase and bolus were repeated every ten minutes. This intended to produce a nearly continuous increase in the propofol plasmatic concentration following intubation. Once SR (as displayed by the SedLine monitor) reached a value between 10% and 30% and was stable for 10 consecutive minutes, the propofol infusion was interrupted and the pig was allowed to recover. Continuous clinical monitoring was performed until the animal was deemed fully awake (animal standing without help, responding to acoustic and visual stimuli, with cardiorespiratory parameters back to baseline).

For data analysis, the time span between the end of the stabilization period (15 min after intubation) and the end of propofol infusion was considered. As propofol was stopped according to SR, the infusion duration and its maximal rate were different between individuals, while a similar DoA was reached. For each pig, the duration of propofol administration was individually divided into 5 equal time intervals (TI1 to TI5). With this transformation, it was assumed that DoA was increasing across comparable levels between pigs from TI1 to TI5. For each TI and for each pig, the mean of the NWRt values was calculated. Baseline values were obtained by averaging the values between 5 and 4 min before the start of propofol administration.

During general anesthesia, heart rate, respiratory rate, rectal temperature, oxygen saturation of hemoglobin (SpO2), invasive blood pressure and end-tidal carbon dioxide (EtCO2) were continuously monitored.

### Statistic

Statistical analysis was performed using the software SigmaPlot Version 15 (Systat Software Inc., San Jose, CA, USA). Normality was assessed using the Shapiro–Wilk test. Results of the NWRt values are presented as median and interquartile ranges (IQR; 25%, 75%). A linear regression between NWRt and TI was performed for each pig. Moreover, baseline NWRt and slopes of the linear regression (b1) were compared among days using a Friedman Repeated Measures Analysis of Variance on Ranks. The statistical significance level was set at *p* < 0.05. Relative NWRt increase compared to baseline was calculated for each TI.

## 3. Results

All the pigs completed the experiment uneventfully. Data from one pig (Pig 2) on one day (Day 1) were excluded due to an incorrect setting ion the NWR device. Placement of the pigs in the slings allowed for easy and straightforward data collection.

In all instances, a minimum of four increments in the propofol dose were necessary to attain the intended SR. Additionally, in seven cases (Pig2-Day3, Pig3-Day1–2–3, Pig4-Day1–3, Pig5-Day3) supplementary elevations in the propofol infusion rate were necessary. The mean (± standard deviation) duration was 77.4 (± 16.9) minutes for the data collection periods and 14.4 (± 5.1) minutes for the time intervals.

A graphical representation of the correlation among the NWRt and the TI for each pig is reported in Figure 3.

No significant difference among days for the baseline NWRt and the b1 values were found.

The median and interquartile ranges of the NWRt values for each TI are reported in Table 1. Their graphical representation and the relative increase compared to the baseline, together with representative NWR recordings and graphical representations of the NWRt for each pig, TI and day, are reported in Appendix A.

## 4. Discussion

In the present study, a rise in the NWRt was found for increasing doses of propofol. Moreover, when the experiment was repeated on different days, no significant differences were found in the baseline values and the slope of increase. This suggests that this method can complement the depth of anesthesia assessment in pigs, at least when the protocol applied in the present study is used.

An NWRt of 7.2 (4.9, 10.5) mA was reached at baseline. No signs of distress were present during the assessment, showing the high compliance and feasibility of this technique in awake animals. To the authors’ knowledge, this is the first study in which NWRt values have been obtained in awake pigs.

Values recorded immediately after intubation were not considered for statistical analysis due to the large difference in propofol dose needed among animals. During that phase, the NWRt reached values of 30.9 (20.1, 34.1) mA. Two possible mechanisms could explain the observed rise. The first and most plausible is of pharmacokinetic (PK)–pharmacodynamic (PD) nature. Indeed, high propofol doses were needed for intubation (mean dose of 14 ± 4 mg/kg); moreover, a rather fast administration was performed. This could have led to an intense and rapid spinal activity depression, mirrored by the NWRt peak, as supported by previous literature [27,28,29]. Another possible reason could be the activation of the diffuse noxious inhibitory control (DNIC) by intubation. The DNIC is the inhibition of activity in convergent or wide dynamic range (WDR)-type nociceptive spinal neurons, triggered by a second, spatially remote noxious stimulus [30,31]. The intubation maneuver, including a mechanical stimulation of the larynx, could have activated the DNIC, diminishing spinal excitability in response to the concurrent electrical stimulations. This could have led to an increase in the NWRt. However, DNIC has been shown to be suppressed during anesthesia in rats [13]. It is not known if the dose of propofol administered was sufficient to reduce or suppress it, and further research should be conducted to clarify this.

Following intubation, we waited for fifteen minutes to allow the stabilization of the propofol plasmatic concentration. As expected, a large NWRt decrease was observed at TI1 (16.9 (13.3, 22.1) mA), presumably linked to the large volume of distribution and fast clearance of propofol in pigs [27].

Finally, before 10–30% suppression ratio was reached, an NWRt increase was observed until TI5.

At TI5, values similar to those recorded at intubation were achieved (29.4 (21.8, 35.3) mA). A ceiling effect of propofol on the NWRt when administered alone or a similar PKPD relationship at both TIs may explain these similar values. Pharmacokinetic analyses should be performed before drawing conclusions.

Similar results were obtained during anesthetic events carried out over separate days. This is in line with data obtained in humans, in which a good NWRt test–retest reliability has been demonstrated [19,20].

That propofol modulates the NWRt is not surprising. The inhibition of movement in response to noxious stimulation is a well-reported property of general anesthetics, with the most probable site of action within the spinal cord [32]. However, it remains unclear what the role of antinociception is in this phenomenon [33,34]. An increase in the NWRt induced by propofol has been reported in humans, and the main mechanistic target has been hypothesized to be at the dorsal horn of the spinal cord [35,36,37]. The present investigation did not provide further insights on the mechanism of action but showed that propofol modulates the NWRt in a dose-dependent manner at clinically relevant anesthetic levels.

The present research has some limitations. First, the fifteen anesthetic events evaluated were recorded from five animals, while data were collected from three separate days per pig. Second, the propofol administration does not reflect the clinical practice, and different results could be obtained with the use of concomitant drugs. Further research should be performed to clarify the effects that different anesthetics and analgesics have on the NWRt. Third, different propofol doses were administered among pigs for the intubation. Even if a 15 min waiting period was allowed to enable the propofol plasmatic concentration to stabilize, no pharmacokinetic analysis was performed to confirm homogeneity among pigs. Fourth, four males and only one female were used for the study. Even if no clear influence of the sex on the NWRt has been reported so far [38,39], it cannot be excluded.

## 5. Conclusions

In the present study, increases in the propofol dose were reflected by increases in the NWRt, and similar results were obtained when the same assessment was carried out on three separate days. Our results suggest that the NWRt can complement the anesthetic level evaluation in pigs receiving propofol. Further investigations should be performed to better characterize the influence that the propofol administration rate and concomitant use of other anesthetic and analgesic drugs might have on this parameter.

## Figures and Tables

**Figure 1 animals-14-01081-f001:**
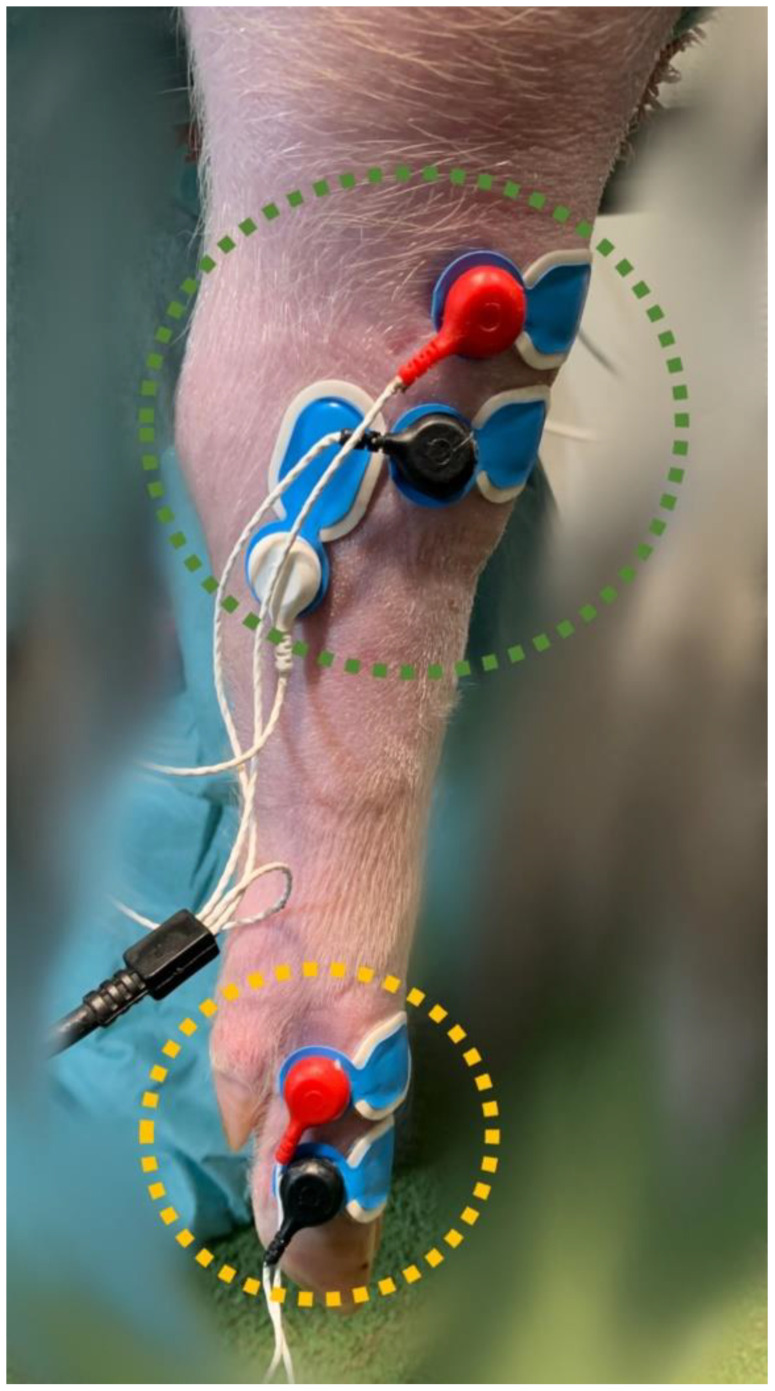
Stimulation (yellow circle) and recording (green circle) surface electrodes, placed on the right hindlimb at the level of the common dorsal digital nerve and the tibialis cranialis muscle, respectively.

**Figure 2 animals-14-01081-f002:**
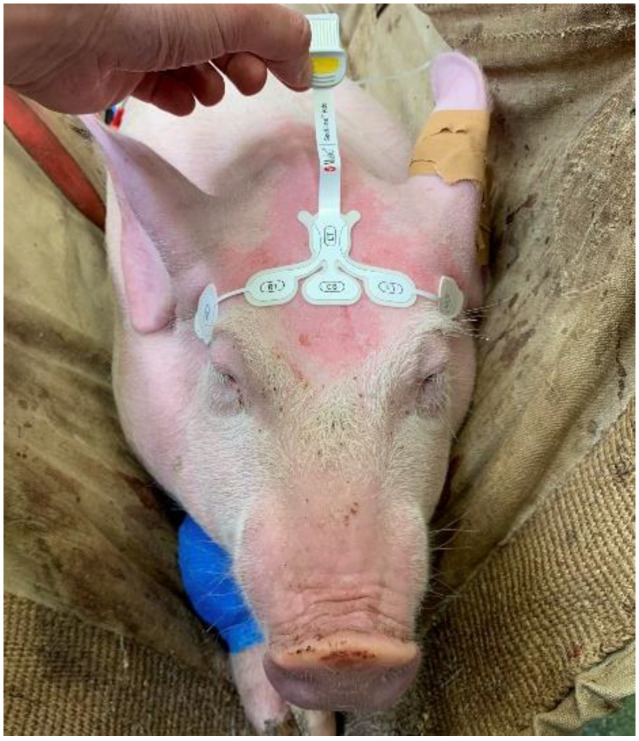
Positioning of the paediatric RD SedLine electroencephalographic sensor in a pig. The electrodes line was placed over the frontal bone, between the eyes, keeping the rostral border of the electrodes on an imaginary line running between the lateral canthi of the eyes. The central GB (ground) and the caudal CT (reference) electrodes were placed on the mid-sagittal line.

**Figure 3 animals-14-01081-f003:**
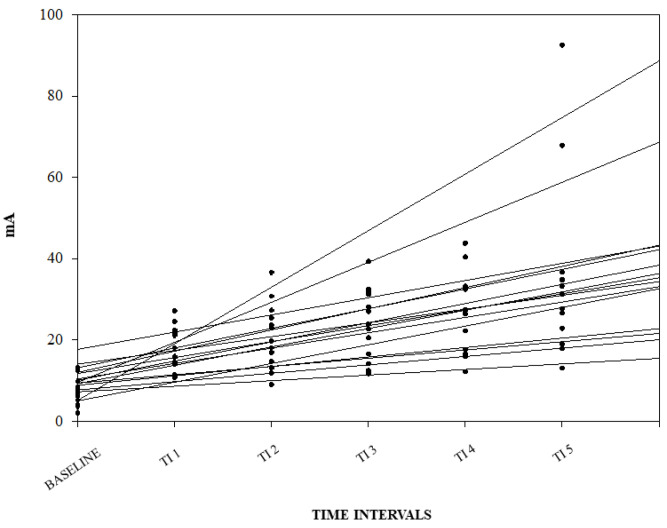
Linear regression between nociceptive withdrawal reflex threshold (NWRt) and time interval (TI) for each pig. Results from the three different days are shown (n = 14; data from one pig missing).

**Table 1 animals-14-01081-t001:** Values of the nociceptive withdrawal reflex threshold (NWRt) at baseline, at the different time intervals (TIs) considered, and at intubation.

	NWRt (mA)		
	Median	IQ 25%	IQ 75%
Baseline	7.2	4.9	10.5
TI1	16.9	13.3	22.1
TI2	18.8	14.2	25.8
TI3	23.2	15.9	31.4
TI4	26.7	17.3	32.9
TI5	29.4	21.8	35.3
Intubation	30.9	20.1	34.1

## Data Availability

The data presented in this study are available in Zenodo at https://doi.org/10.5281/zenodo.8354916, accessed on 18 September 2023.

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
