# Peer review of "How Is the Nociceptive Withdrawal Reflex Influenced by Increasing Doses of Propofol in Pigs?"

_animals, 2024, doi:10.3390/ani14071081_

Round 1
Reviewer 1 Report
Comments and Suggestions for Authors
Reviewer comments on Manuscript number: animals-2878958
Dear authors,
The manuscript shows the effects of increasing doses of propofol on the noceiceptive withdrawal reflex in pigs.
The manuscript is well written and the results are interesting, as a scarce information about this topic is available in pigs.
Some few detailes have to be adressed before the review process continues.
In material and methods.
L74. Please specify each point of the ARRIVE guideline followed in the study.
L108. Please specify which peripheral vein and artery were catheterized.
L 134. Please specify the equipment used to monitor the EEG.
In the supplementary material, it would be desirable to identify each table and what is showed there.
Please include the possible clinical relevance of your study.
Thank you
Author Response
Reviewer 1
Dear authors,
The manuscript shows the effects of increasing doses of propofol on the noceiceptive withdrawal reflex in pigs. The manuscript is well written and the results are interesting, as a scarce information about this topic is available in pigs. Some few detailes have to be adressed before the review process continues.
L74. Please specify each point of the ARRIVE guideline followed in the study.
Thank you for your comment. To our understanding, all the points have been reported (see image in the attached file).
What has not been included are the points 4 and 5 (not applicable), since the study did not compare groups (all the pigs underwent the same protocol, and each animal underwent the same procedure on the three different days), and no blinding could be applied (all the animal received one single drug, that was propofol).
If more information is needed, we will be happy to include it.
L108. Please specify which peripheral vein and artery were catheterized.
The information is now reported in Line 110-111.
L 134. Please specify the equipment used to monitor the EEG.
The information has been added in the text at line 139.
In the supplementary material, it would be desirable to identify each table and what is showed there.
Thank you for noticing it. The information has been added.
Please include the possible clinical relevance of your study.
Thank you, a sentence has been added in the Discussion (227-229).

Reviewer 2 Report
Comments and Suggestions for Authors
I think that the data and its analysis are useful and that this article will be suitable for publication in Animals. I have no suggestions for improvement , just a few questions listed below:
Page 2, Line 96: Only one day of acclimatization? Its very short and just for curiosity, is this the normal routines at the university of Bern?
Page 4 Line 123: 50% randomisation. What was randomized, the intensity? Any thoughts about sensitisation in case of mixed intensities?
The study have some limitations concerning the dose and lack of PK analysis of Propofol. However this is stated and discussed at the end of the paper.
A very interesting study. I hope there will be more to read in the future with PK data included.
Author Response
Reviewer 2
I think that the data and its analysis are useful and that this article will be suitable for publication in Animals. I have no suggestions for improvement , just a few questions listed below:
Page 2, Line 96: Only one day of acclimatization? Its very short and just for curiosity, is this the normal routines at the university of Bern?
We recognize that one day could be too short (not a standard procedure at our Institution, but in this case chosen for organizational reasons). In the present study, the acclimatization of one day was performed before the placement of the subcutaneous venous port, but at least 4 days (in total: 1+3) were waited until starting the main experiment. A sentence has been added in the text (lines 96-99)
Page 4 Line 123: 50% randomisation. What was randomized, the intensity? Any thoughts about sensitisation in case of mixed intensities?
The randomization was done on the stimulation interval (added in the text). This is done automatically by the device to avoid habituation.
The study have some limitations concerning the dose and lack of PK analysis of Propofol. However this is stated and discussed at the end of the paper.
A very interesting study. I hope there will be more to read in the future with PK data included.
Reviewer 3 Report
Comments and Suggestions for Authors
Dear authors.
I have carefully reviewed the current manuscript. The aim of your study was to observe and describe how increasing doses of propofol affect the NWRt in pigs. Considering the lack of means to assess the depth of anesthesia or the level of uncosciousness objectively, your study should provide useful information concerning depth of anesthesia assessment or even identification of the appropriate surgical anesthetic depth.
However, I think you should clarify some details related to the study design before this manuscript is accepted for publication.
Introduction
I really liked this section. It is tight, concise and to the point. Maybe you could provide some more information about MAC determination and its utilization.
Materials and Methods
Lines 80-81: So, you utilized 5 pigs for 15 treatments? Was that a rationale design?
Lines 81-82: Was the washout period of 3 days enough?
Lines 89-90: I am a little confused. Which 3 groups are you referring to? The treatment was the same, and the pigs were 5 in total. Which was the reason for anesthetizing each pig for 3 times, apart from increasing your sample size, if the treatment was the same?
What was the randomization plan, if there was any? How was the order the same 5 pigs received the same treatments, decided?
Lines 156-158: Which was the equilibration period after each increase?
Lines 166-168: How long were the "equal time intervals''? Which was the association between time intervals (TI1 - TI5) and time of each propofol infusion increases? Was every time interval correlated to the preset propofol infusion increase? A timeline would be useful.
Results
Which was the duration of each experiment? Was it different between each treatment? In Lines 190-191 you mention that supplementary elevations of the propofol infusion rate were needed. Didn't it affect the total general anesthesia time?
Author Response
Reviewer 3
Dear authors.
I have carefully reviewed the current manuscript. The aim of your study was to observe and describe how increasing doses of propofol affect the NWRt in pigs. Considering the lack of means to assess the depth of anesthesia or the level of uncosciousness objectively, your study should provide useful information concerning depth of anesthesia assessment or even identification of the appropriate surgical anesthetic depth. However, I think you should clarify some details related to the study design before this manuscript is accepted for publication.
Introduction
I really liked this section. It is tight, concise and to the point. Maybe you could provide some more information about MAC determination and its utilization.
Thank you for your comment. This point was also previously discussed among the authors. However, since the paper is not focused on MAC determination, its argumentation was not extended. Indeed, MAC introduction is only done to inform the reader on the most common way to assess depth of anesthesia in animals, but this is not linked to our methods/results.
Materials and Methods
Lines 80-81: So, you utilized 5 pigs for 15 treatments? Was that a rationale design?
Thank you for your question. The design was made for another (main) study (lines 76-79); thus, it was not thought for the analysis performed in this manuscript. Since the main study foresaw that the 5 animals had to undergo the same anesthetic procedure (sole propofol) for 3 times (having different treatments=methylphenidate administered only at its end), we collected the data on the 3 different days. This way, we finally had data from 15 separate anesthetic events.
Lines 81-82: Was the washout period of 3 days enough?
Thank you for asking. Due to the fast metabolization of propofol in pigs and the fact that it was the only anesthetic drug administered, we believe the washout period was sufficient. We also think that this period was sufficient for eliminate the treatment drug administered at the end of anesthesia for the main study (methylphenidate; not reported in the present manuscript).
Lines 89-90: I am a little confused. Which 3 groups are you referring to? The treatment was the same, and the pigs were 5 in total. Which was the reason for anesthetizing each pig for 3 times, apart from increasing your sample size, if the treatment was the same?
As mentioned in the previous comment, the main study foresaw the administration of 3 different treatments (3 different doses of methylphenidate; 3 groups) at the end of anaesthesia (data not included in the present paper). Thus, the sample size was based on this outcome.
Each animal received one of the three treatments (methylphenidate) only at the end of the anesthesia (that was always the same, namely with the sole propofol). Thus, for the present study, we only analyzed data collected during anesthesia (before the treatments were injected). This allowed us to have a total of 15 anesthetic events (since each of the 5 animals underwent anesthesia for 3 times).
What was the randomization plan, if there was any? How was the order the same 5 pigs received the same treatments, decided?
The randomization was performed for the main study, but this did not influence the present research, since each animal underwent the same anesthetic protocol (sole propofol) before the administration of one of the three treatments (methylphenidate; main study).
Lines 156-158: Which was the equilibration period after each increase?
It was 10 minutes (reported in lines 160-161).
Lines 166-168: How long were the "equal time intervals''? Which was the association between time intervals (TI1 - TI5) and time of each propofol infusion increases? Was every time interval correlated to the preset propofol infusion increase? A timeline would be useful.
Each time interval was different for each pig. In particular, the time between 15 minutes after intubation (to allow propofol stabilization after intubation; let’s call it “point A”) and the stop of propofol infusion (stopped when the target was reached; let’s call it “point B”) was considered. The length between the two points (point B – point A) was always different, since each pig needed a different amount of propofol/time to reach the target (suppression ratio between 10 and 30%). For this reason, we divided this time span in 5 equal time points (e.g., if one pig needed 50 minutes to go from point A to point B, each time interval would have been of 10 minutes; if one pig needed 80 minutes to go from point A to point B, each time interval would have been of 16 minutes).
Thus, the recording was not linked anymore to the single propofol increases. It is indeed important to notice that the step-increase of the anesthetic was performed to mirror a linear increase of the plasmatic concentration (as reported in lines 158-159). Doing a comparison between “step increases” among pigs would have not made much sense, since at the first increase some pigs were mildly sedated, while others were already in a deep sedation status.
I provide in the attached file a small graph to explain it better.
Among pigs, the mean time of “point B-point A” was 77.4 ± 16.9 minutes, while the mean time-interval (TI) was 14.4 ± 5.1 minutes. This information has been added in the text.
Results
Which was the duration of each experiment? Was it different between each treatment? In Lines 190-191 you mention that supplementary elevations of the propofol infusion rate were needed. Didn't it affect the total general anesthesia time?
The mean total duration of the whole experiment (from propofol start to propofol stop) was 104.4 ± 16.4 minutes. Mean (± standard deviation) duration was 77.4 (± 16.9) minutes for the data collection periods, and 14.4 (± 5.1) minutes for the time intervals. The latters have been added in the text (lines 199-201).
We did not assess if there was a difference among treatments since they were administered only at the end of propofol anesthesia (thus outside the window of data collection).
Concerning the supplementary elevations of propofol infusion: yes, they affected the total anesthesia duration (each elevation was done every ten minutes). For this reason, we decided not to compare the values at each increase but to divide the data in equal time points (see also answers above). This allowed us to have comparable data among pigs.

Round 2
Reviewer 3 Report
Comments and Suggestions for Authors
Dear authors.
I have no other comments.